# REGULARIZING CNNS WITH LOCALLY CONSTRAINED DECORRELATIONS

**Pau Rodríguez**[†]**, Jordi Gonzàlez**[†,‡]**, Guillem Cucurull**[†]**, Josep M. Gonfaus**[‡]**, Xavier Roca**[†,‡]
[†]Computer Vision Center - Univ. Autònoma de Barcelona (UAB), 08193 Bellaterra, Catalonia Spain
[‡]Visual Tagging Services, Campus UAB, 08193 Bellaterra, Catalonia Spain

## ABSTRACT

Regularization is key for deep learning since it allows training more complex models while keeping lower levels of overfitting. However, the most prevalent regularizations do not leverage all the capacity of the models since they rely on reducing the effective number of parameters. Feature decorrelation is an alternative for using the full capacity of the models but the overfitting reduction margins are too narrow given the overhead it introduces. In this paper, we show that regularizing negatively correlated features is an obstacle for effective decorrelation and present OrthoReg, a novel regularization technique that locally enforces feature orthogonality. As a result, imposing locality constraints in feature decorrelation removes interferences between negatively correlated feature weights, allowing the regularizer to reach higher decorrelation bounds, and reducing the overfitting more effectively. In particular, we show that the models regularized with OrthoReg have higher accuracy bounds even when batch normalization and dropout are present. Moreover, since our regularization is directly performed on the weights, it is especially suitable for fully convolutional neural networks, where the weight space is constant compared to the feature map space. As a result, we are able to reduce the overfitting of state-of-the-art CNNs on CIFAR-10, CIFAR-100, and SVHN.

## 1 INTRODUCTION

Neural networks perform really well in numerous tasks even when initialized randomly and trained with Stochastic Gradient Descent (SGD) (see Krizhevsky et al. (2012)). Deeper models, like Googlenet (Szegedy et al. (2015)) and Deep Residual Networks (Szegedy et al. (2015); He et al. (2015a)) are released each year, providing impressive results and even surpassing human performances in well-known datasets such as the Imagenet (Russakovsky et al. (2015)). This would not have been possible without the help of regularization and initialization techniques which solve the overfitting and convergence problems that are usually caused by data scarcity and the growth of the architectures.

From the literature, two different regularization strategies can be defined. The first ones consist in reducing the complexity of the model by (i) reducing the effective number of parameters with weight decay (Nowlan & Hinton (1992)), and (ii) randomly dropping activations with Dropout (Srivastava et al. (2014)) or dropping weights with DropConnect (Wan et al. (2013)) so as to prevent feature co-adaptation. Due to their nature, although this set of strategies have proved to be very effective, they do not leverage all the capacity of the models they regularize.

The second group of regularizations is those which improve the effectiveness and generality of the trained model without reducing its capacity. In this second group, the most relevant approaches decorrelate the weights or feature maps, e.g. Bengio & Bergstra (2009) introduced a new criterion so as to learn slow decorrelated features while pre-training models. In the same line Bao et al. (2013) presented "incoherent training", a regularizer for reducing the decorrelation of the network activations or feature maps in the context of speech recognition. Although regularizations in the second group are promising and have already been used to reduce the overfitting in different tasks, even with the presence of Dropout (as shown by Cogswell et al. (2016)), they are seldom used in the large scale image recognition domain because of the small improvement margins they provide together with the computational overhead they introduce.

|            | **Base**      | **DeCov**     | **OrthoReg** |
|------------|---------------|---------------|--------------|
| **MLP**    | $8.1\ 10^8$   | $5.2\ 10^{10}$| $9.7\ 10^9$  |
| **ResNet-110** | $6.5\ 10^{10}$ | $3.4\ 10^{14}$ | $3.4\ 10^8$ |

Table 1: Count of the Flops for the models used in this paper: the 3-hidden-layer MLP and the 110-layer ResNet we use later in the experiments section when not regularized, using DeCov (Cogswell et al. (2016)) and using OrthoReg. Batch size is set to 128, the same we use to train the ResNet. Regularizing weights is orders of magnitude faster than regularizing activations.

Although they are not directly presented as regularizers, there are other strategies to reduce the over-fitting such as Batch Normalization (Ioffe & Szegedy (2015)), which decreases the overfitting by reducing the internal covariance shift. In the same line, initialization strategies such as "Xavier" (Glorot & Bengio (2010)) or "He" (He et al. (2015b)), also keep the same variance at both input and output of the layers in order to preserve propagated signals in deep neural networks. Orthogonal initialization techniques are another family which set the weights in a decorrelated initial state so as to condition the network training to converge into better representations. For instance, Mishkin & Matas (2016) propose to initialize the network with decorrelated features using orthonormal initialization (Saxe et al. (2013)) while normalizing the variance of the outputs as well.

In this work we hypothesize that regularizing negatively correlated features is an obstacle for achieving better results and we introduce OrhoReg, a novel regularization technique that addresses the performance margin issue by only regularizing positively correlated feature weights. Moreover, OrthoReg is computationally efficient since it only regularizes the feature weights, which makes it very suitable for the latest CNN models. We verify our hypothesis through a series of experiments: first using MNIST as a proof of concept, secondly we regularize wide residual networks on CIFAR-10, CIFAR-100, and SVHN (Netzer et al. (2011)) achieving the lowest error rates in the dataset to the best of our knowledge.

## 2 DEALING WITH WEIGHT REDUNDANCIES

Deep Neural Networks (DNN) are very expressive models which can usually have millions of parameters. However, with limited data, they tend to overfit. There is an abundant number of techniques in order to deal with this problem, from L1 and L2 regularizations (Nowlan & Hinton (1992)), early-stopping, Dropout or DropConnect. Models presenting high levels of overfitting usually have a lot of redundancy in their feature weights, capturing similar patterns with slight differences which usually correspond to noise in the training data. A particular case where this is evident is in AlexNet (Krizhevsky et al. (2012)), which presents very similar convolution filters and even "dead" ones, as it was remarked by Zeiler & Fergus (2014).

In fact, given a set of parameters $\theta_{I,j}$ connecting a set of inputs $I = \{i_1, i_2, \ldots, i_n\}$ to a neuron $h_j$, two neurons $\{h_j, h_k\}$, $j \neq k$ will be positively correlated, and thus fire always together if $\theta_{I,j} = \theta_{I,k}$ and negatively correlated if $\theta_{I,j} = -\theta_{I,k}$. In other words, two neurons with the same or slightly different weights will produce very similar outputs. In order to reduce the redundancy present in the network parameters, one should maximize the amount of information encoded by each neuron. From an information theory point of view, this means one should not be able to predict the output of a neuron given the output of the rest of the neurons of the layer. However, this measure requires batch statistics, huge joint probability tables, and it would have a high computational cost.

In this paper, we will focus on the weights correlation rather than activation independence since it still is an open problem in many neural network models and it can be addressed without introducing too much overhead, see Table 1. Then, we show that models generalize better when different feature detectors are enforced to be dissimilar. Although it might seem contradictory, CNNs can benefit from having repeated filter weights with different biases, as shown by Li et al. (2016). However, those repeated filters must be shared copies and adding too many unshared filter weights to CNNs increases overfitting and the need for stronger regularization (Zagoruyko & Komodakis (May 2016)). Thus, our proposed method and multi-bias neural networks are complementary since they jointly increase the representation power of the network with fewer parameters.

In order to find a good target to optimize so as to reduce the correlation between weights, it is first required to find a metric to measure it. In this paper, we propose to use the cosine similarity between feature detectors to express how strong is their relationship. Note that the cosine similarity is equivalent to the Pearson correlation for mean-centered normalized vectors, but we will use the term correlation for the sake of clarity.

## 2.1 Orthogonal weight regularization

This section introduces the orthogonal weight regularization, a regularization technique that aims to reduce feature detector correlation enforcing local orthogonality between all pairs of weight vectors. In order to keep the magnitudes of the detectors unaffected, we have chosen the cosine similarity between the vector pairs in order to solely focus on the vectors angle $\beta \in [-\pi, \pi]$. Then, given any pair of feature vectors of the same size $\theta_1, \theta_2$ the cosine of their relative angle is:

$$\cos(\theta_1, \theta_2) = \frac{\langle \theta_1, \theta_2 \rangle}{||\theta_1||||\theta_2||} \tag{1}$$

Where $\langle \theta_1, \theta_2 \rangle$ denotes the inner product between $\theta_1$ and $\theta_2$. We then square the cosine similarity in order to define a regularization cost function for steepest descent that has its local minima when vectors are orthogonal:

$$C(\theta) = \frac{1}{2} \sum_{i=1}^{n} \sum_{j=1, j \neq i}^{n} \cos^2(\theta_i, \theta_j) = \frac{1}{2} \sum_{i=1}^{n} \sum_{j=1, j \neq i}^{n} \left( \frac{\langle \theta_i, \theta_j \rangle}{||\theta_i||||\theta_j||} \right)^2 \tag{2}$$

Where $\theta_i$ are the weights connecting the output of the layer $l - 1$ to the neuron $i$ of the layer $l$, which has $n$ hidden units. Interestingly, minimizing this cost function relates to the minimization of the Frobenius norm of the cross-covariance matrix without the diagonal. This cost will be added to the global cost of the model $J(\theta; X, y)$, where $X$ are the inputs and $y$ are the labels or targets, obtaining $\tilde{J}(\theta; X, y) = J(\theta; X, y) + \gamma C(\theta)$. Note that $\gamma$ is an hyperparameter that weights the relative contribution of the regularization term. We can now define the gradient with respect to the parameters:

$$\frac{\delta}{\delta \theta_{(i,j)}} C(\theta) = \sum_{k=1, k \neq i}^{n} \frac{\theta_{(k,j)} \langle \theta_i, \theta_k \rangle}{\langle \theta_i, \theta_i \rangle \langle \theta_k, \theta_k \rangle} - \frac{\theta_{(i,j)} \langle \theta_i, \theta_k \rangle^2}{\langle \theta_i, \theta_i \rangle^2 \langle \theta_k, \theta_k \rangle} \tag{3}$$

The second term is introduced by the magnitude normalization. As magnitudes are not relevant for the vector angle problem, this equation can be simplified just by assuming normalized feature detectors:

$$\frac{\delta}{\delta \theta_{(i,j)}} C(\theta) = \sum_{k=1, k \neq i}^{n} \theta_{(k,j)} \langle \theta_i, \theta_k \rangle \tag{4}$$

We then add eq. 4 to the backpropagation gradient:

$$\Delta \theta_{(i,j)} = -\alpha \left( \nabla J_{\theta_{(i,j)}} + \gamma \sum_{k=1, k \neq i}^{n} \theta_{(k,j)} \langle \theta_i, \theta_k \rangle \right) \tag{5}$$

Where $\alpha$ is the global learning rate coefficient, $J$ any target loss function for the backpropagation algorithm.

Although this update can be done sequentially for each feature-detector pair, it can be vectorized to speedup computations. Let $\Theta$ be a matrix where each row is a feature detector $\theta_{(I,j)}$ corresponding to the normalized weights connecting the whole input $I$ of the layer to the neuron $j$. Then, $\Theta \Theta^t$ contains the inner product of each pair of vectors $i$ and $j$ in each position $i, j$. Subsequently, we

---

**Algorithm 1** Orthogonal Regularization Step.

---

**Require:** Layer parameter matrices $\Theta^l$, regularization coefficient $\gamma$, global learning rate $\alpha$.
1: **for** each layer $l$ to regularize **do**
2: $\eta_1 = norm\_rows(\Theta^l)$ ▷ Keep norm of the rows of $\Theta^l$.
3: $\Theta_1^l = div\_rows(\Theta^l, \eta_1)$ ▷ Keep a $\Theta_1^l$ with normalized rows.
4: $innerProdMat = \Theta_1^l transpose(\Theta_1^l)$
5: $\nabla\Theta_1^l = \gamma(innerProdMat - diag(innerProdMat))\Theta_1^l$ ▷ Second term in eq. 6
6: $\Delta\Theta^l = -\alpha(\nabla J_{\Theta^l} + \gamma\nabla\Theta_1^l)$ ▷ Complete eq. 6.
7: **end for**

---

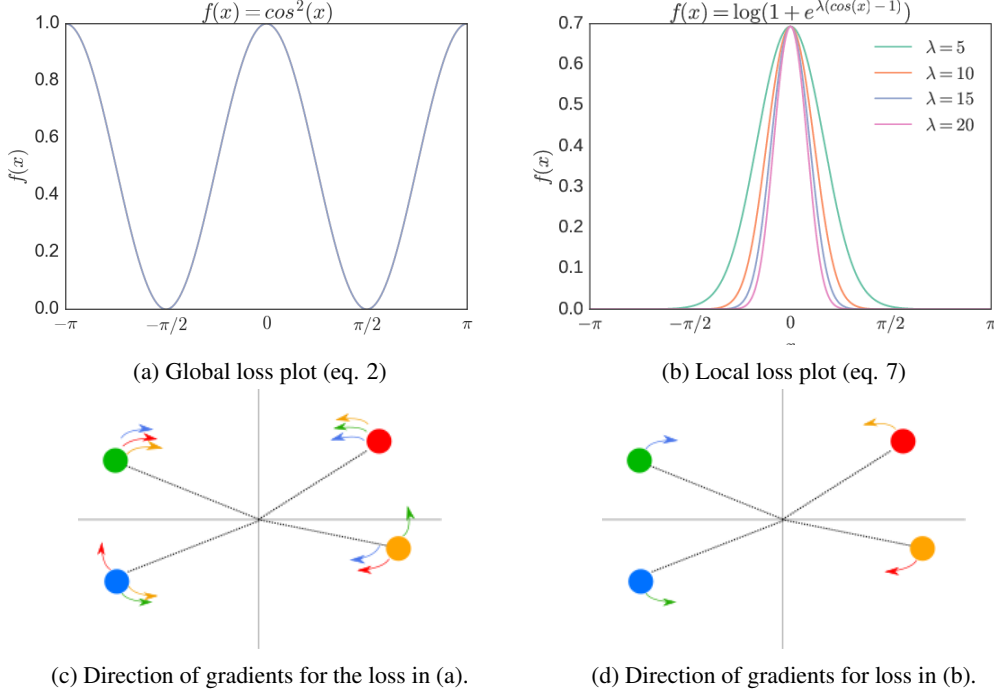

(a) Global loss plot (eq. 2) (b) Local loss plot (eq. 7)

(c) Direction of gradients for the loss in (a). (d) Direction of gradients for loss in (b).

Figure 1: Comparison between the two loss functions represented by eq.2 and 7. (a) is the original loss, (b) is the new loss that discards negative correlations given for different $\lambda$ values. It can be seen $\lambda = 10$ reaches a plateau when approximating to $\frac{\pi}{2}$. (c) and (d) shows the directions of the gradients for the two loss functions above. For instance, a red arrow coming from a green ball represents the gradient of the loss between the red and green balls with respect to the green one. In (d) most of the arrows disappear since the loss in (b) only applies to angles smaller than $\frac{\pi}{2}$.

subtract the diagonal so as to ignore the angle from each feature with respect to itself and multiply by $\Theta$ to compute the final value corresponding to the sum in eq. 5:

$$\Delta\Theta = -\alpha\Big(\nabla J_\Theta + \gamma(\Theta\Theta^t - diag(\Theta\Theta^t))\Theta\Big) \tag{6}$$

Where the second term is $\nabla C_\Theta$. Algorithm 1 summarizes the steps in order to apply OrthoReg.

## 2.2 NEGATIVE CORRELATIONS

Note that the presented algorithm, based on the cosine similarity, penalizes any kind of correlation between all pairs of feature detectors, i.e. the positive and the negative correlations, see Figure 1a. However, negative correlations are related to inhibitory connections, competitive learning, and self-organization. In fact, there is evidence that negative correlations can help a neural population to increase the signal-to-noise ratio (Chelaru & Dragoi (2016)) in the V1. In order to find out

the advantages of keeping negative correlations, we propose to use an exponential to squash the gradients for angles greater than $\frac{\pi}{2}$ ($orthogonal$):

$$C(\theta) = \sum_{i=1}^{n} \sum_{j=1,j\neq i}^{n} \log(1 + e^{\lambda(cos(\theta_i,\theta_j)-1)}) = \log(1 + e^{\lambda(\langle\theta_i,\theta_j\rangle-1)}), \ ||\theta_i|| = ||\theta_j|| = 1 \quad (7)$$

Where $\lambda$ is a coefficient that controls the minimum angle-of-influence of the regularizer, i.e. the minimum angle between two feature weights so that there exists a gradient pushing them apart, see Figure 1b. We empirically found that the regularizer worked well for $\lambda = 10$, see Figure 2b. Note that when $\lambda \simeq 10$ the loss and the gradients approximate to zero when vectors are at more than $\frac{\pi}{2}$ (orthogonal). As a result of incorporating the squashing function on the cosine similarity, negatively correlated feature weights will not be regularized. This is different from all previous approaches and the loss presented in eq. 2, where all pairs of weight vectors influence each other. Thus, from now on, the loss in eq. 2 is named as global loss and the loss in eq. 7 is named as local loss.

The derivative of eq. 7 is:

$$\frac{\delta}{\delta\theta_{(i,j)}}C(\theta) = \sum_{k=1,k\neq i}^{n} \lambda\frac{e^{\lambda\langle\theta_i,\theta_k\rangle}\theta_{(k,j)}}{e^{\lambda\langle\theta_i,\theta_k\rangle} + e^{\lambda}} \quad (8)$$

Then, given the element-wise exponential operator $\exp$, we define the following expression in order to simplify the formulas:

$$\hat{\Theta} = \exp(\lambda(\Theta\Theta^t)) \quad (9)$$

and thus, the $\Delta$ in vectorial form can be formulated as:

$$\nabla C_\Theta = \lambda\frac{(\hat{\Theta} - diag(\hat{\Theta}))\Theta}{\hat{\Theta} - diag(\hat{\Theta}) + e^{\lambda}} \quad (10)$$

In order to provide a visual example, we have created a $2D$ toy dataset and used the previous equations for positive and negative $\gamma$ values, see Figure 2. As expected, it can be seen that the angle between all pairs of adjacent feature weights becomes more uniform after regularization. Note that Figure 2b shows that regularization with the global loss (eq. 2) results in less uniform angles than using the local loss as shown in 2c (which corresponds to the local loss presented in eq. 7) because vectors in opposite quadrants still influence each other. This is why in Figure 2d, it can be seen that the mean nearest neighbor angle using the global loss (b) is more unstable than the local loss (c). As a proof of concept, we also performed gradient ascent, which minimizes the angle between the vectors. Thus, in Figures 2e and 2f, it can be seen that the locality introduced by the local loss reaches a stable configuration where feature weights with angle $\frac{\pi}{2}$ are too far to attract each other.

The effects of global and local regularizations on Alexnet, VGG-16 and a 50-layer ResNet are shown on Figure 3. As it can be seen, OrthoReg reaches higher decorrelation bounds. Lower decorrelation peaks are still observed when the input dimensionality of the layers is smaller than the output since all vectors cannot be orthogonal at the same time. In this case, local regularization largely outperforms global regularization since it removes interferences caused by negatively correlated feature weights. This suggests why increasing fully connected layers' size has not improved networks performance.

## 3 EXPERIMENTS

In this section we provide a set of experiments that verify that (i) training with the proposed regularization increases the performance of naive unregularized models, (ii) negatively correlated feature weights are useful, and (iii) the proposed regularization improves the performance of state-of-the-art models.

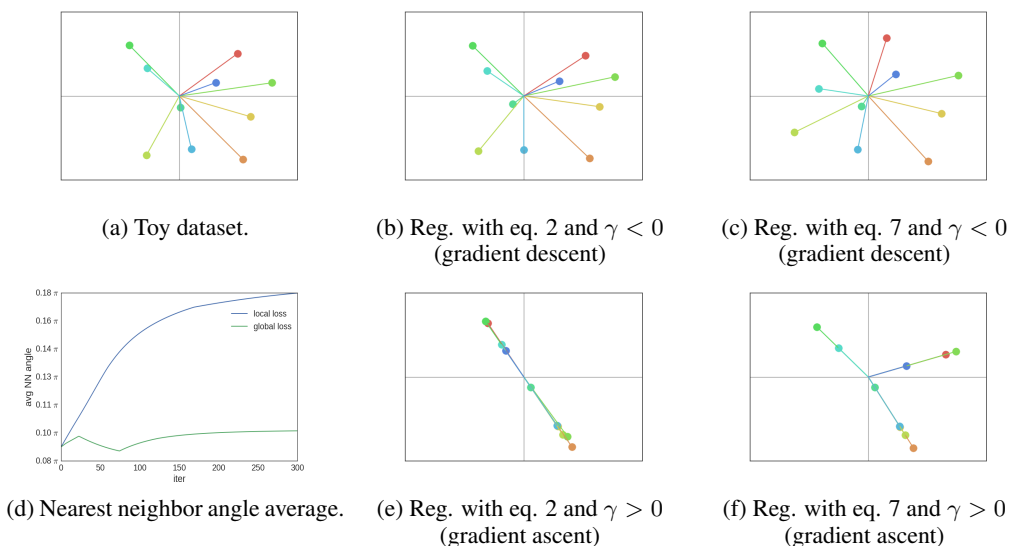

(a) Toy dataset.

(b) Reg. with eq. 2 and $\gamma < 0$ (gradient descent)

(c) Reg. with eq. 7 and $\gamma < 0$ (gradient descent)

(d) Nearest neighbor angle average.

(e) Reg. with eq. 2 and $\gamma > 0$ (gradient ascent)

(f) Reg. with eq. 7 and $\gamma > 0$ (gradient ascent)

Figure 2: 2D toy dataset regularized with global loss (eq. 2) and local loss (eq. 7). (a) shows the initial 2D randomly generated dataset. (b) the dataset after 300 regularization steps using the global loss and (c) using the local loss. (d) the evolution of the mean nearest neighbor angle for the global loss (b) and the local loss (c). (e) and (f) correspond to (b) and (c) but using gradient ascent instead of gradient descent as a sanity-check.

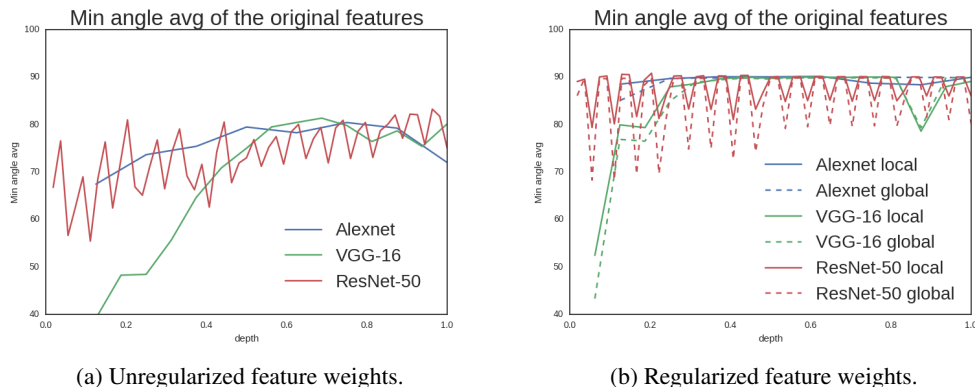

(a) Unregularized feature weights.

(b) Regularized feature weights.

Figure 3: Effects of local and global regularization on the Alexnet, VGG-16 and 50-layer-ResNet weights. The regularized versions reach higher decorrelation bounds (in terms of minimum angle) than the unregularized counterparts.

## 3.1 VERIFICATION EXPERIMENTS

As a sanity check, we first train a three-hidden-layer Multi-Layer Perceptron (MLP) with `ReLU` non-liniarities on the MNIST dataset (LeCun et al. (1998)). Our code is based in the `train-a-digit-classifier` example included in `torch/demos`[1], which uses an upsampled version of the dataset ($32 \times 32$). The only pre-processing applied to the data is a global standardization. The model is trained with SGD and a batch size of 200 during 200 epochs. No momentum neither weight decay was applied. By default, the magnitude of the weights of this experiments is recovered after each regularization step in order to prove the regularization only affects their angle.

**Sensitivity to hyperparameters.** We train a three-hidden-layer MLP with 1024 hidden units, and different $\gamma$ and $\lambda$ values so as to verify how they affect the performance of the model. Figure 4a

[1]`https://github.com/torch/demos`

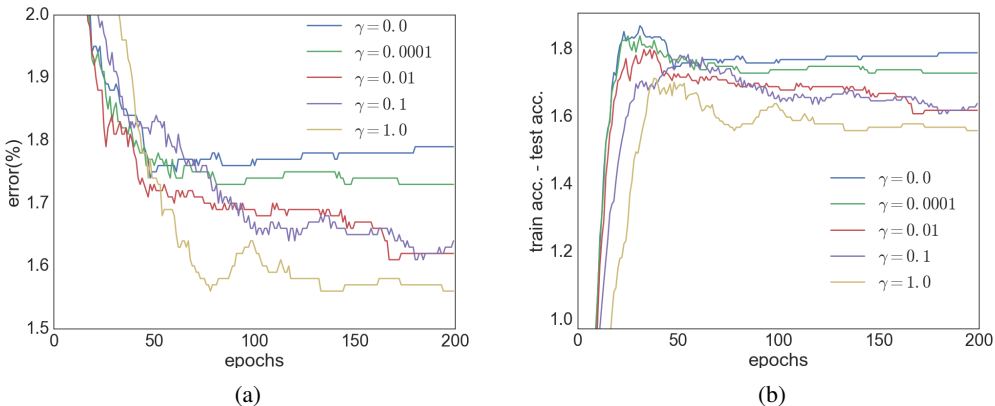

(a) (b)

Figure 4: (a) The evolution of the error rate on the MNIST validation set for different regularization magnitudes. It can be seen that for $\gamma = 1$ it reaches the best error rate $(1.45\%)$ while the unregularized counterpart $(\gamma = 0)$ is $1.74\%$. (b) Measures the overfitting of the model for different $\gamma$, confirming that higher regularization rates decrease overfitting.

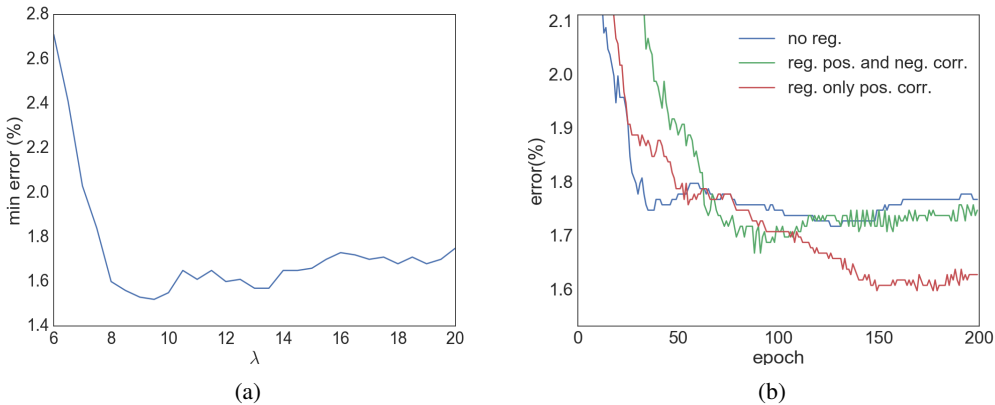

(a) (b)

Figure 5: (a) Shows the minimum error rate for different $\lambda$ values. (b) Classification error on MNIST for different loss functions. Not regularizing negative correlated feature weights (eq. 7) results in better test error than regularizing them (eq.2).

shows that the model effectively achieves the best error rate for the highest gamma value $(\gamma = 1)$, thus proving the advantages of the regularization. On Figure 4b, we verify that higher regularization rates produce more general models. Figure 5a depicts the sensitivity of the model to $\lambda$. As expected, the best value is found when lambda corresponds to Orthogonality $(\lambda \simeq 10)$.

**Negative Correlations.** Figure 5b highlights the difference between regularizing with the global or the local regularizer. Although both regularizations reach better error rates than the unregularized counterpart, the local regularization is better than the global. This confirms the hypothesis that negative correlations are useful and thus, performance decreases when we reduce them.

**Compatibility with initialization and dropout.** To demonstrate the proposed regularization can help even when other regularizations are present, we trained a CNN with (i) dropout (`c32-c64-l512-d0.5-l10`)[2] or (ii) LSUV initialization (Mishkin & Matas (2016)). In Table 2, we show that best results are obtained when orthogonal regularization is present. The results are consistent with the hypothesis that OrthoReg, as well as Dropout and LSUV, focuses on reducing the model redundancy. Thus, when one of them is present, the margin of improvement for the others is reduced.

---

[2]$cxx$ = convolution with $xx$ filters. $lxx$ = fully-connected with $xx$ units. $dxx$ = dropout with prob $xx$.

| OrthoReg | Base | Base+Dropout | Base+LSUV |
|---|---|---|---|
| None | 0.92 | $0.70 \pm 0.01$ | 0.86 |
| Conv Layers | 0.75 | $0.69 \pm 0.03$ | 0.83 |
| All Layers | **0.75** | $\mathbf{0.66 \pm 0.03}$ | **0.79** |

Table 2: Error rates for a small CNN trained with the MNIST dataset. OrthoReg leads to much better results when no other improvements such as Dropout and LSUV are present but it can still make small accuracy increments when these two techniques are present.

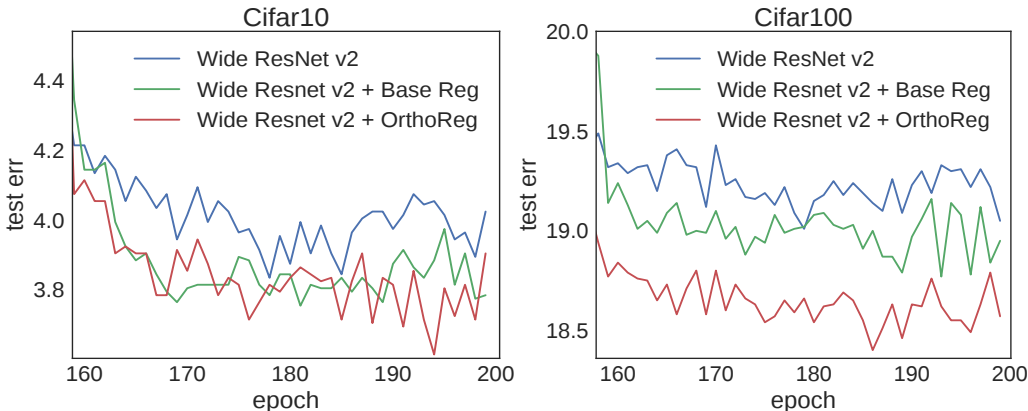

Figure 6: Wide ResNet error rate on Cifar10 and Cifar100. OrthoReg shows better performance than the base regularizer (all feature maps), and the unregularized counterparts.

## 3.2 REGULARIZATION ON CIFAR-10 AND CIFAR-100

We show that the proposed OrthoReg can help to improve the performance of state-of-the-art models such as deep residual networks (He et al. (2015a)). In order to show the regularization is suitable for deep CNNs, we successfuly regularize a 110-layer ResNet[3] on CIFAR-10, decreasing its error from 6.55% to 6.29% without data augmentation.

In order to compare with the most recent state-of-the-art, we train a wide residual network (Zagoruyko & Komodakis (November 2016)) on CIFAR-10 and CIFAR-100. The experiment is based on a `torch` implementation of the 28-layer and 10th width factor wide deep residual model, for which the median error rate on CIFAR-10 is 3.89% and 18.85% on CIFAR-100[4]. As it can be seen in Figure 6, regularizing with OrthoReg yields the best test error rates compared to the baselines.

The regularization coefficient $\gamma$ was chosen using grid search although similar values were found for all the experiments, specially if regularization gradients are normalized before adding them to the weights. The regularization was equally applied to all the convolution layers of the (wide) ResNet. We found that, although the regularized models were already using weight decay, dropout, and batch normalization, best error rates were always achieved with OrthoReg.

Table 3 compares the performance of the regularized models with other state-of-the-art results. As it can be seen the regularized model surpasses the state of the art, with a 5.1% relative error improvement on CIFAR-10, and a 1.5% relative error improvement on CIFAR-100.

## 3.3 REGULARIZATION ON SVHN

For SVHN we follow the procedure depicted in Zagoruyko & Komodakis (May 2016), training a wide residual network of `depth=28`, `width=4`, and dropout. Results are shown in Table 4. As it

---

[3]https://github.com/gcr/torch-residual-networks
[4]https://github.com/szagoruyko/wide-residual-networks

| Network | CIFAR-10 | CIFAR-100 | Augmented |
|---|---|---|---|
| Maxout (Goodfellow et al. (2013)) | 9.38 | 38.57 | YES |
| NiN (Lin et al. (2014)) | 8.81 | 35.68 | YES |
| DSN (Lee et al. (2015)) | 7.97 | 34.57 | YES |
| Highway Network (Srivastava et al. (2015)) | 7.60 | 32.24 | YES |
| All-CNN (Springenberg et al. (2015)) | 7.25 | 33.71 | NO |
| 110-Layer ResNet (He et al. (2015a)) | 6.61 | 28.4 | NO |
| ELU-Network (Clevert et al. (2016)) | 6.55 | **24.28** | NO |
| **OrthoReg on 110-Layer ResNet**\* | **6.29** $\pm 0.19$ | $28.33 \pm 0.5$ | NO |
| LSUV (Mishkin & Matas (2016)) | 5.84 | - | YES |
| Fract. Max-Pooling (Graham (2014)) | 4.50 | 27.62 | YES |
| Wide ResNet v1 (Zagoruyko & Komodakis (May 2016))\* | 4.37 | 20.40 | YES |
| **OrthoReg on Wide ResNet v1 (May 2016)**\* | $4.32 \pm 0.05$ | $19.50 \pm 0.03$ | YES |
| Wide ResNet v2 (Zagoruyko & Komodakis (November 2016))\* | 3.89 | 18.85 | YES |
| **OrthoReg on Wide ResNet v2 (November 2016)**\* | $\mathbf{3.69 \pm 0.01}$ | $\mathbf{18.56 \pm 0.12}$ | YES |

Table 3: Comparison with other CNNs on CIFAR-10 and CIFAR-100 (Test error %). Orthogonally regularized residual networks achieve the best results to the best of our knowldege. Only single-crop results are reported for fairness of comparison. \*Median over 5 runs as reported by Zagoruyko & Komodakis (November 2016).

| Model | Error rate |
|---|---|
| NiN (Lin et al. (2014)) | 2.35 |
| DSN (Lee et al. (2015)) | 1.92 |
| Stochastic Depth ResNet (Huang et al. (2016)) | 1.75 |
| Wide Resnet (Zagoruyko & Komodakis (May 2016)) | 1.64 |
| **OrthoReg on Wide Resnet** | **1.54** |

Table 4: Comparison with other CNNs on SVHN. Wide Resnets regularized with OrthoReg show better performance.

can be seen, we reduce the error rate from $1.64\%$ to $1.54\%$, which is the lowest value reported on this dataset to the best of our knowledge.

## 4 DISCUSSION

Regularization by feature decorrelation can reduce Neural Networks overfitting even in the presence of other kinds of regularizations. However, especially when the number of feature detectors is higher than the input dimensionality, its decorrelation capacity is limited due to the effects of negatively correlated features. We showed that imposing locality constraints in feature decorrelation removes interferences between negatively correlated feature weights, allowing regularizers to reach higher decorrelation bounds, and reducing the overfitting more effectively.

In particular, we show that the models regularized with the constrained regularization present lower overfitting even when batch normalization and dropout are present. Moreover, since our regularization is directly performed on the weights, it is especially suitable for fully convolutional neural networks, where the weight space is constant compared to the feature map space. As a result, we are able to reduce the overfitting of 110-layer ResNets and wide ResNets on CIFAR-10, CIFAR-100, and SVHN improving their performance. Note that despite OrthoReg consistently improves state of the art ReLU networks, the choice of the activation function could affect regularizers like the one presented in this work. In this sense, the effect of asymmetrical activations on feature correlations and regularizers should be further investigated in the future.

ACKNOWLEDGEMENTS

Authors acknowledge the support of the Spanish project TIN2015-65464-R (MINECO/FEDER), the 2016FI_B 01163 grant of Generalitat de Catalunya, and the COST Action IC1307 iV&L Net (European Network on Integrating Vision and Language) supported by COST (European Cooperation in Science and Technology). We also gratefully acknowledge the support of NVIDIA Corporation with the donation of a Tesla K40 GPU and a GTX TITAN GPU, used for this research.

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
