# Peer review of "Regularizing CNNs with Locally Constrained Decorrelations"

_ICLR 2017 — accepted_

[Reviewer Comment · AnonReviewer2 · 02 Dec 2016]
**Very good results**
soundness 5 · clarity 5

This is a good paper with the appropriate experimentation. Regularization must be tested on deep and complex topologies, near to state of the art. Other papers test reg. with simple models where regularization helps but are not in the edge...

[Official Review · AnonReviewer1 · rating 7 · confidence 4 · 15 Dec 2016]
**Orthogonality of weight features might not always lead to improved performance**
originality 5

Encouraging orthogonality in weight features has been reported useful for deep networks in many previous works. The authors present a explicit regularization cost to achieve de-correlation among weight features in a layer and encourage orthogonality. Further, they also show why and how negative correlations can and should be avoided for better de-correlation. 

Orthogonal weight features achieve better generalization in case of large number of trainable parameters and less training data, which usually results in over-fitting. As also mentioned by the authors biases help in de-correlation of feature responses even in the presence of correlated features (weights). Regularization techniques like OrthoReg can be more helpful in training deeper and leaner networks, where the representational capacity of each layer is low, and also generalize better.

Although the improvement in performances is not significant the direction of research and the observations made are promising.

[Official Review · AnonReviewer2 · rating 7 · confidence 3 · 16 Dec 2016]
**Good results on deep nets**
soundness 5 · clarity 5

The author proposed a simple but yet effective technique in order to regularized neural networks. The results obtained are quite good and the technique shows to be effective when it it applied even on state of the art topologies, that is welcome because some regularization techniques used to be applied in easy task or on a initial configuration which results are still far from the best known results.

[Official Review · AnonReviewer3 · rating 7 · confidence 4 · 16 Dec 2016]
**Solid experimental validation**
originality 4

The paper proposes a new regulariser for CNNs that penalises positive correlations between feature weights, but does not affect negative correlations. An alternative version which penalises all correlations regardless of sign is also considered. The paper refers to these as "local" and "global" respectively, which I find a bit confusing as these are very general terms that can mean a plethora of things.

The experimental validation is quite rigorous. Several experiments are conducted on benchmark datasets (MNIST, CIFAR-10, CIFAR-100, SVHN) and improvements are demonstrated in most cases. While these improvements may seem modest, the baselines are already very competitive as the authors pointed out. In some cases it does raise some questions about statistical significance though. More results with the global regulariser (i.e. not just on MNIST) would have been interesting, as the main novelty in the paper seems to be leaving the negative correlations alone, so it would be interesting to see exactly how much of a difference this makes.

One of my main concerns is ambiguity stemming from the fact that the paper sometimes discusses activations and sometimes filter weights, but refers to both as "features". However, the authors have already said they will address this.

The paper somewhat ignores interactions with the choice of nonlinearity, which seems like it could be very important; especially because the goal is to obtain feature activations that are uncorrelated, and this is done only by applying a penalty to the weights (i.e. in a data-agnostic way and also ignoring any nonlinearity). I believe the authors already mentioned in their responses to reviewer questions that this would be addressed, but I think this important and it definitely needs to be discussed.

In response to the authors' answer to my question about the role of biases: as they point out, it is perfectly possible to combine their proposed technique with the "multi-bias" approach, but this was not really my point. Rather, the latter is an example that challenges the idea that features should not be positively correlated / redundant, which seems to be the assumption that this work is built upon. My current intuition is that it's okay to have correlated features, as long as you're not wasting model capacity on them. This is the case for "multi-bias", seeing as the weights are shared across sets of correlated features.

The dichotomy between regularisation methods that reduce capacity and those that don't which is described in the introduction seems a bit arbitrary to me, especially considering that weight decay is counted among the former and the proposed method is counted among the latter. I think this very much depends on ones definition of model capacity (clearly weight decay does not actually reduce the number of parameters in a model).

Overall, the work is perhaps a bit incremental, but it seems to be well-executed. The results are convincing, even if they aren't particularly ground-breaking.

[Public Comment · Pau Rodriguez · 13 Jan 2017]
**Paper v2.0**

We thank again the reviewers for the thorough revision and the valuable feedback. We have updated the paper accordingly:

1. We have applied OrthoReg on the new version of Wide ResNets, consistently reducing the error rates to 3.69%, and 18.56% on Cifar10, and Cifar100 respectively.

2.  Figure 6 has been updated so as to compare the base regularization to the regularization of positively correlated feature weights. All the curves are now obtained from Wide ResNet v2.

Multi-bias neural networks are now discussed in the third paragraph of section 2.
3. The need for investigating the effects of activation functions like ReLU on the different existing regularizations is now reflected on the discussion section.

4. When appearing alone, the word “features” has been replaced for “feature weights” or “feature activations” accordingly.
The dichotomy between regularization types has been changed according to reviewer 3 comments.

[Final Decision · Program Chairs · 06 Feb 2017]
**ICLR committee final decision**

The paper presents a new regularization approach for deep learning that penalizes positive correlations between features in a network. The experimental evaluation is solid, and suggests the proposed regularized may help in learning better convolutional networks (but the gains are relatively small).